# Micrometastases in Sentinel Lymph Nodes Represent a Significant Negative Prognostic Factor in Early-Stage Cervical Cancer: A Single-Institutional Retrospective Cohort Study

**DOI:** 10.3390/cancers12061438

**Published:** 2020-05-31

**Authors:** Roman Kocian, Jiri Slama, Daniela Fischerova, Anna Germanova, Andrea Burgetova, Ladislav Dusek, Pavel Dundr, Kristyna Nemejcova, Jiri Jarkovsky, Silvie Sebestova, Filip Fruhauf, Lukas Dostalek, Tereza Ballaschova, David Cibula

**Affiliations:** 1Gynecologic Oncology Center, Department of Obstetrics and Gynecology, First Faculty of Medicine, Charles University in Prague and General University Hospital in Prague, 128 00 Prague, Czech Republic; kocianroman@seznam.cz (R.K.); Jiri.Slama@vfn.cz (J.S.); daniela.fischerova@seznam.cz (D.F.); ena.german@gmail.com (A.G.); fruhauffilip@centrum.cz (F.F.); Lukas.Dostalek@vfn.cz (L.D.); Tereza.Ballaschova@vfn.cz (T.B.); 2Department of Radiology, First Faculty of Medicine, Charles University in Prague and General University Hospital in Prague, 128 00 Prague, Czech Republic; Andrea.Burgetova@vfn.cz; 3Institute of Biostatistics and Analyses, Faculty of Medicine, Masaryk University, 625 00 Brno, Czech Republic; dusek@iba.muni.cz (L.D.); jarkovsky@iba.muni.cz (J.J.); 4Institute of Health Information and Statistics of the Czech Republic, 128 01 Prague, Czech Republic; Zouharova@iba.muni.cz; 5Department of Pathology, First Faculty of Medicine, Charles University in Prague and General University Hospital in Prague, 128 00 Prague, Czech Republic; pavel.dundr@vfn.cz (P.D.); kristyna.nemejcova@vfn.cz (K.N.)

**Keywords:** micrometastasis, isolated tumor cells, sentinel lymph node, cervical cancer, pathological ultrastaging, prognostic parameters, risk of recurrence

## Abstract

The data on the prognostic significance of low volume metastases in lymph nodes (LN) are inconsistent. The aim of this study was to retrospectively analyze the outcome of a large group of patients treated with sentinel lymph node (SLN) biopsy at a single referral center. Patients with cervical cancer, stage T1a-T2b, common tumor types, negative LN on preoperative staging, treated by primary surgery between 01/2007 and 12/2016, with at least unilateral SLN detection were included. Patients with abandoned radical surgery due to intraoperative SLN positivity detected by frozen section were excluded. All SLNs were postoperatively processed by an intensive protocol for pathological ultrastaging. Altogether, 226 patients were analyzed. Positive LN were detected in 38 (17%) cases; macrometastases (MAC), micrometastases (MIC), isolated tumor cells (ITC) in 14, 16, and 8 patients. With the median follow-up of 65 months, 22 recurrences occurred. Disease-free survival (DFS) reached 90% in the whole group, 93% in LN-negative cases, 89% in cases with MAC, 69% with MIC, and 87% with ITC. The presence of MIC in SLN was associated with significantly decreased DFS and OS. Patients with MIC and MAC should be managed similarly, and SLN ultrastaging should become an integral part of the management of patients with early-stage cervical cancer.

## 1. Introduction

One of the main controversies in the management of cervical cancer is currently the uncertain prognostic importance of micrometastases (MIC) and isolated tumor cells (ITC). These small metastatic lesions, by definition ≤2 mm, were in the past reported extremely rarely by a standard pathological assessment of pelvic lymph nodes (LN). As the acceptance and popularity of SLN has increased, and pathological processing is much more intensive in SLN than in other pelvic LN, and about 10% to 15% of patients with early-stage tumors are detected with MIC or ITC in their SLN [1,2,3,4,5,6,7,8,9,10,11,12,13,14,15,16]. Available data on the impact of MIC or ITC for prognosis are not consistent [1,14,15,16,17,18,19,20]. Since the risk of recurrence is very low in early-stage cervical cancer, any assessment of prognostic importance of MIC requires large cohorts.

The controversy about the prognostic significance of MIC and ITC has major consequences for clinicians, who must decide if they should manage these cases as LN-positive, and this is even more important for pathologists. If small-size metastases showed no impact on the outcome of patients, the intention of SLN examination would be limited to the detection of macrometastases above 2 mm in size. In such cases, a protocol for SLN assessment would be much less time-consuming, less expensive, and would not include immunohistochemistry. An intraoperative one-step nucleic acid amplification method (OSNA) has been recently proposed as an alternative method to ultrastaging [21].

The aim of the study was to retrospectively analyze data from a large cohort of patients treated in a tertiary gynecologic oncology center, where SLN biopsy has been used in the management of cervical cancer since 2004, and where a standardized intensive protocol for SLN pathologic ultrastaging has been applied since 2009.

## 2. Results

In total, 226 patients were included in the analysis (Table 1).

The majority had squamous cell cancer (76%), non-fertility-sparing surgical procedure (88%), and stage pT1b1 (69%). Parametrial invasion was reported by pathology in nine cases; in six cases, an initial invasion was known before surgery, and three were upstaged based on the pathology report (cT1b → pT2b). Out of those six cases treated by primary surgery, two were symptomatic cases with severe bleeding from the tumor, and four patients with initial parametrial invasion diagnosed by imaging preferred surgery over primary chemoradiation. Radical parametrectomy (radical hysterectomy or radical trachelectomy) type C1 or C2 was performed in 47% and 36% of patients, respectively.

Overall SLN detection rate reached 93% at least on one side of the pelvis; bilateral detection rate was achieved in 80% of all cases and was comparable in subgroups with tumors <2 cm, 2–3.9 cm, and ≥4 cm (79%, 83%, 76%) [22]. There were two patients with negative SLN and macrometastases in non-SLN from systematic pelvic lymphadenectomy; the false negativity of sentinel lymph node ultrastaging reached 1% only (Table 2). 

Lymph node involvement was diagnosed in 38 cases (17%), including MAC in 14, MIC in 16, and ITC in 8 cases. Adjuvant radiotherapy or chemoradiation was given to 37 cases (16%), due to LN involvement (27 cases), positive vaginal margin (2), and parametrial involvement (8). Amongst patients with positive LNs, adjuvant treatment was not received by 4/14 cases with MAC (three T1b1 cases who rejected radiotherapy; one case with early cervical progression after fertility-sparing treatment), 2/16 cases with MIC (one patient rejected radiotherapy; one case was treated in an early period when MIC was not considered an indication for adjuvant treatment in the absence of other prognostic risk factors), and 5/8 cases with ITC (all five cases treated in an earlier period when ITC had not been considered an indication for adjuvant treatment in the absence of other prognostic risk factors).

With the median follow-up of 65 months, 22 recurrences occurred: eight in the pelvis only, four in distant sites only, and ten combined. Six recurrences developed in patients after fertility-sparing treatment. Three cervical (2) or pelvic (1) recurrences after abdominal radical trachelectomy were salvaged by further treatment. Three pelvic (2) or combined (1) recurrences after conization were fatal (Table 3).

Patient 17, who had a 25 mm tumor and deep stromal invasion (tumor free distance TFD = 0), refused both radical trachelectomy and adjuvant radiotherapy. Amongst patients with positive LNs who did not receive adjuvant treatment (11), only one with ITC developed recurrence. Disease-free survival (DFS) with the median follow-up of 65 months reached 90% in the whole group, 93% in LN-negative patients, 89% in patients with MAC, 69% in patients with MIC, and 87% with ITC. DFS was significantly worse in cases with MAC (*p* = 0.037) and MIC (*p* = 0.001) in comparison to LN-negative cases (Figure 1). 

Similarly, OS was significantly worse in groups with MAC (*p* < 0.001) and MIC (*p* < 0.001) in comparison to LN-negative patients (Figure 2). 

Both DFS (*p* = 0.717) and OS (*p* = 0.839) were similar in patients with MAC and MIC. Parameters significant for the risk of recurrence by the univariate analysis included adenosquamous tumor type (HR = 5.08; *p* = 0.032), presence of LVSI (HR = 2.95; *p* = 0.018), number of positive LNs (HR = 1.5; *p* = 0.015), LN positivity (MAC or MIC) (HR = 4.03; *p* = 0.002), MAC in LN (HR = 3.61; *p* = 0.046), MIC in LN (HR = 4.62; *p* = 0.004), TFD binarized (cut-off value ≤3.5 mm) (HR = 9.0; *p* = 0.033), tumor size binarized (cut-off value >33.5 mm) (HR = 2.56; *p* = 0.029), and adjuvant treatment (HR = 3.46; *p* = 0.005) (Table 4). 

None of the parameters significant in univariate analysis remained significant in the multivariate model (Table 5).

## 3. Discussion

In a large retrospective cohort of patients from a single institution, the presence of MIC in SLN was a significant independent negative prognostic factor. Patients with MAC and MIC had significantly and similarly decreased DFS and OS in comparison to LN-negative patients.

Numerous papers on SLN in cervical cancer presented data on the prevalence of SLN [11,12,13,23,24], and some suggested an association of MIC with other traditional tumor-related risk factors such as tumor size or LVSI [15,16,25]. Very few papers, however, evaluated the impact of MIC on the prognosis and the data are varying (Table 6).

For the first time, the potential significance of MIC was suggested by the French group (Marchiole 2005) [15]. In a case-control study, they compared a group of 26 recurred patients with the same number of matched controls without recurrence. A hysterectomy specimen was reassessed for LVSI; LN samples were serial-sectioned and stained using cytokeratin. The relative risk of recurrence was 2.44 (*p* < 0.01) for MIC and 2.64 (*p* < 0.01) for LVSI. In a Brazilian study, all pelvic LNs from 289 patients in stages IB–IIA were reassessed, finding 11 cases with MIC (3.8%) and 37 cases with MAC (12.8%) (Fregnani 2006) [17]. The low prevalence of MIC corresponded to a very low intensity of LN pathological processing. With the median follow-up of 8.5 years, 43 recurrences (15%) occurred. The presence of MIC was a significant independent prognostic factor (HR = 3.2; 95% CI: 1.1–9.6) with five-year DFS at 89%, 80%, and 50% in patients with N0, MIC, and MAC, respectively. In 2008, a German group presented the outcome of a large group of 894 patients with IB–IIB cervical cancer. They re-examined samples from positive LN, measuring the size of metastases, using original slides without any further processing (Horn 2008) [14]. Five-year DFS was significantly lower in both groups with MAC (62%) and MIC (69%) in comparison to those with negative LN (87%). In the largest retrospective study published so far, data from 645 cases were collected from seven institutions (Cibula 2012) [1]. All patients had SLN biopsy followed by pelvic lymph node dissection, and SLNs were processed by pathological ultrastaging. Both MAC and MIC were associated with similar and significantly decreased overall survival (MAC: HR = 6.85; 95% CI: 2.59–18.05; MIC: HR = 6.86; 95% CI: 2.09–22.61). In another multi-institutional retrospective study, tissue blocks were recut and evaluated for the presence of MIC in a group of 129 patients who were LN-negative at the time of primary treatment (Stany 2015) [18]. Any immunoreactive tumor cells were classified as MIC, not distinguishing MIC and ITC. This can explain the high proportion of 26 (20%) patients with MIC detected by re-evaluation. The presence of MIC was not associated with a negative outcome. There were, however, only 11 recurrences in this group (8.5%), and patients with MIC were more likely to receive adjuvant radiotherapy than those with negative LN (39% vs. 18%). In a similar study, LN tissue was reviewed and stained by immunohistochemistry in a group of 83 LN-negative patients. The presence of MIC was the strongest independent predictor of the recurrence by multivariate analysis (OR = 11.73; 95%CI: 1.57–87.8; *p* = 0.017), outweighing all traditional tumor-related variables such as LVSI, stromal invasion, or tumor size (Colturado, 2016) [19]. Recently, data from the prospective French study SENTICOL were analyzed for the presence and impact of MIC and ITC (Guani 2019) [20]. All LNs from 139 patients were reprocessed, although the protocol for ultrastaging of that many hundreds of LNs is not fully described. Positive LNs were found in 25 patients (18%), including eight cases with only MIC and eight cases with only ITC. Since 14 cases with MIC or ITC were reported in the original report, it seems that two more cases were identified by an additional pathological review of non-SLNs (Bats 2012) [26]. With the median follow-up of 36 months, only 13 (9%) recurrences occurred. Surprisingly, all types of LN metastases, MAC as well as MIC or ITC, were associated with a decreased survival. 

There are multiple reasons which can explain discrepant results in the literature. Firstly, the risk of recurrence is low in the early stages of cervical cancer, usually around 10%, so it requires a large cohort to demonstrate any significant impact on the prognosis. Even in our study, which is, to our knowledge, the largest single institutional retrospective cohort, we reported only 22 (10%) recurrences with the medium survival of more than five years. Secondly, and more importantly, in the absence of any universal protocol for SLN ultrastaging, pathological processing is so different that such discrepancies inevitably impact the accuracy of detection of not only MIC but also small MAC [27]. Other than this paper, only two out of nine previously published studies evaluating the impact of MIC on prognosis included pathological ultrastaging of SLN (Cibula 2012, Guani 2019) [1,20]. Thirdly, the designs of the studies differed considerably. In some of them, SLN was prospectively assessed by ultrastaging, while other pelvic LNs were processed by standard H&E evaluation. In others, tissue blocks from all pelvic LNs were re-evaluated retrospectively from patients who were LN-negative at the time of the treatment. There are substantial differences in cohort sizes (49–894), in disease stages, and in the proportion of cases who received adjuvant therapy (0–33%). Notably, only two papers reported an adjuvant radiotherapy rate in cases with MIC.

The prevalence of cases with LN involvement varies widely in cohorts of patients treated by primary surgery; from 7% to 20% [1,8,28,29,30,31,32,33,34,35]. This is mostly due to selection criteria for primary surgical treatment. The occurrence of 6% of MAC cases in our study is rather low, taking into account that it entailed the whole spectrum of early stages, including 24% of tumors larger than 4 cm. The low rate of LN positivity could be explained by the intraoperative triage of patients based on SLN status. Radical surgery was abandoned if LN involvement was detected intraoperatively by frozen section, and these cases were excluded from our analysis.

The strengths of our study include the uniform management of patients throughout the study period in a single center, the large cohort size, and the intensive protocol of SLN ultrastaging.

Only a small proportion of patients received adjuvant treatment (16%). It should be emphasized that adjuvant (chemo)radiation was administered to a similar proportion of cases with MAC and MIC, so the outcome was not impacted by a different postoperative management. Although this study is the largest single institutional cohort evaluating the impact of MIC on prognosis, the main limitation remains the small number of patients with MIC and the limited number of recurrences. Due to the excellent outcome of current early-stage cervical cancer management and the relatively low prevalence of MIC, thousands of patients would be required for a prospective study powered to address the impact on survival. Even the two ongoing prospective trials on SLN in cervical cancer patients (SENTIX, NCT02494063; SENTICOL III, NCT03386734) are not designed to bring the evidence. This applies even more to ITC. Other than the prevalence of ITC, which usually equals to half of MIC, an additional reason lies in unreliable pathological detection. SLN ultrastaging cannot be intensive enough to detect all ITCs, so their prevalence will always be only relative, reflecting the intensity of the protocol for SLN assessment [1,31,36,37].

## 4. Materials and Methods 

### 4.1. Methods

This study was approved by the Ethics Committee of the General University Hospital in Prague (project No. 1587/17 S-IV). Patients with cervical cancer stage pT1a–pT2b (squamous cell cancers, adenocarcinomas or adenosquamous cancer), without enlarged or suspicious pelvic LN on preoperative imaging, treated by primary surgery with curative intent, with at least unilateral SLN in the pelvis detected, and treated in one tertiary center between January 2007 and December 2016, were included in the study. Excluded were patients in whom radical hysterectomy or fertility-sparing treatment was abandoned due to intraoperative detection of positive SLN, patients with rare tumor types, patients in whom SLN was not detected at least on one pelvic side, and patients who received neoadjuvant chemotherapy.

A combined technique with both radioactive tracer (99Tc, long protocol, application 12 h before surgery, 4 × 20 MBq) and blue dye (application at the beginning of the surgery, 2 mL not diluted or diluted in 2 mL of saline) was used for SLN detection either by laparoscopy or by laparotomy. In small tumors, both tracers were applied superficially into the cervical stroma. The syringe was kept in place for a few seconds after application to avoid retrograde leakage of the tracer. The application technique was modified in cases with large tumors, as previously described (application into the residual stroma by a spinal needle, continuous control of vaginal leak when injected into the necrotic tissue) [22]. All well-defined pelvic regions were carefully explored and searched for all blue and/or radioactive lymph nodes with hand-held gamma probe. All identified SLNs were submitted for intraoperative pathologic evaluation. Further radical surgery was abandoned if any type of metastases, gross parametrial invasion, or any distant spread was identified intraoperatively, and such patients were referred for primary chemoradiation instead.

If SLNs were intraoperatively confirmed negative, full pelvic lymph node dissection was completed, except in stage T1a/LVSI-negative patients. Systematic lymphadenectomy included removal of all fatty lymphatic tissue from seven pelvic regions which are well defined by exact anatomical landmarks, i.e., regions with the most frequent occurrence of positive lymph nodes. These involve bilateral obturator fossa, external and common iliac, plus presacral regions. Parametrial lymph nodes were removed together with parametria as part of radical hysterectomy specimen [38].

The type of radical parametrectomy was classified according to the Querleu–Morrow classification system [39,40], and the extent of parametrectomy was tailored to risk factors known preoperatively [41]. Adjuvant radiotherapy or chemoradiation was administered if patients had positive LN, parametrial involvement, or positive vaginal margins.

Patients were followed in the center for at least five years after treatment. Survival data were controlled by matching data with the Czech National Registry of Death.

Pathological tumor stage (pT), tumor type, grading, LVSI status, parametrial involvement, LN involvement (MAC, MIC, ITC), number of positive LN, largest tumor size assessed by ultrasound (US), largest tumor size assessed by pathology (P), minimal tumor-free distance (TFD) assessed by ultrasound (US), depth of stromal invasion (DSI), and tumor volume calculated by the formula for ellipsoid from pathological measurement (P) were evaluated by a univariate analysis. Tumor-free distance (TFD) was defined as the minimal uninvolved stroma between the tumor and pericervical ring (dense hyperechogenic layer on ultrasound and hypointense layer on MRI) on either side of the cervix.

### 4.2. Pathology

At the time of surgery, all submitted SLNs were cut along their longest axis, and both halves of each node were examined with frozen sectioning techniques. SLNs with a diameter of less than 3 mm were not examined by frozen section. After that, SLNs as well as all other pelvic LNs were fixed in 10% formalin, sliced at 2 mm intervals, and embedded in paraffin. SLN ultrastaging protocol consisted of two consecutive sections (4 μm thick) obtained in regular 150 μm intervals at four levels. The first section was stained with H&E, and the second section was examined immunohistochemically with an antibody against cytokeratins (AE1/AE3, 1:50 dilution; Dako, Glostrup, Denmark). The presence of MAC, MIC, and ITC was classified according to the TNM system. Macrometastasis was defined as a metastasis >2 mm in the largest diameter, MIC as a metastasis between 0.2 and 2 mm, and ITC as individual tumor cells or small clusters of cells <0.2 mm in diameter.

### 4.3. Statistics

Standard descriptive statistics were applied in the analysis; absolute and relative frequencies for categorical variables and median supplemented with the 5th–95th percentile range for continuous variables. The influence of patient characteristics on survival was analyzed using univariate and multivariate Cox proportional hazard models and described using hazard ratios (HRs) and their 95% confidence intervals. Cut-off values for continuous variables were determined by ROC analysis; the criterion was the highest value of the sum of sensitivity and specificity. Kaplan–Meier methodology was adopted for the visualization of survival data; the statistical significance of differences in survival curves among groups of patients was tested using the log rank test. Analysis was computed using SPSS 25.0.0.1 (IBM Corporation 2018).

## 5. Conclusions

In conclusion, based on the results of our study and a critical review of the literature, there is growing evidence that the presence of MIC in SLN is associated with significant negative impact on the survival, which is similar to patients with MAC. Despite caveats in current evidence and discrepancies in available data, patients with MIC should be managed with the same criteria as patients with MAC, and SLN biopsy and its ultrastaging should be implemented into routine management. SLN ultrastaging is undoubtedly a substantially more time and cost consuming practice if compared to routine LN assessment; however, it enables the identification of an additional subgroup of around 10% of patients with MIC who would be otherwise missed. If cases with all types of metastases (MAC, MIC, ITC) are excluded, a remaining subgroup of LN negative patients has an excellent prognosis. 

## Figures and Tables

**Figure 1 cancers-12-01438-f001:**
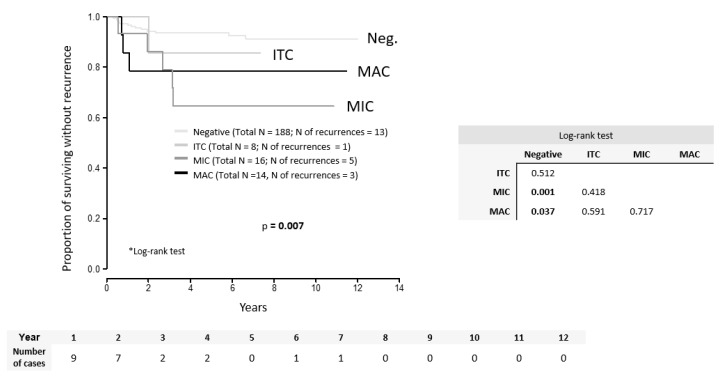
Disease-free survival according to the type of LN involvement. ITC = isolated tumor cells, MIC = micrometastasis, MAC = macrometastasis.

**Figure 2 cancers-12-01438-f002:**
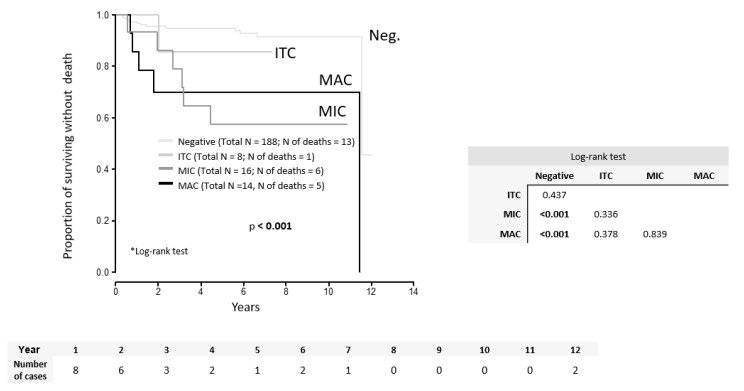
Overall survival according to the type of LN involvement. ITC = isolated tumor cells, MIC = micrometastasis, MAC = macrometastasis.

**Table 1 cancers-12-01438-t001:** Characteristics of the group (*n* = 226).

Characteristic		Whole Cohort ^1^
Age (years)		42.2 (26.2; 67.9)
BMI		24.3 (18.4; 36.2)
Stage pT	1a1	8 (3.5%)
	1a2	7 (3.1%)
	1b1	157 (69.4%)
	1b2	42 (18.6%)
	2a	3 (1.3%)
	2b	9 (4.0%)
Tumor type	Adenocarcinoma	49 (21.7%)
	Adenosquamous	6 (2.7%)
	Squamous	171 (75.7%)
Grade	1	21 (9.3%)
	2	95 (42.0%)
	3	91 (40.3%)
	missing	19 (8.4%)
LVSI		98 (43.4%)
Fertility sparing treatment	Conisation ST RT	11 (4.9%) 4 (1.8%) 12 (5.3%)
Surgical approach	Open Laparoscopic	196 (86.7%) 30 (13.3%)
Type of parametrectomy	A	5 (2.2%)
	B	13 (5.8%)
	C	2 (0.9%)
	C1	106 (46.9%)
	C2	82 (36.3%)
	missing	18 (8.0%)
SLNB	Bilateral Unilateral	196 (86.7%) 30 (13.3%)
Pelvic lymphadenectomy		212 (93.8%)
Number of LN per patient		36.0 (4.0; 59.0)
Type of LN positivity	MAC	14 (6.2%)
	MIC	16 (7.1%)
	ITC	8 (3.5%)
	Negative	188 (83.2%)
Largest tumor size (US) ^2^		25.5 (3.4; 52.0)
Largest tumor size (P) ^3^		26.0 (6.0; 65.0)
Depth of stromal invasion (P) ^3^		15.0 (5.0; 25.0)
Tumor volume (P) ^3^		4336.3 (113.1; 43,987.6)
Adjuvant treatment		37 (16.4%)
	Combined RT	13 (5.8%)
	Chemoradiation	24 (10.6%)
Follow-up length (months)		64.5 (7.0; 123.0)
Time to recurrence (months)		61.5 (6.4; 123.0)
Recurrences		22 (9.7%)
Deaths	DOD DOC	25 (9.7%) 18 (7.9%) 7 (3.1%)

^1^ absolute and relative frequencies for categorical variables; median supplemented with 5th–95th percentile range for continuous variables; ^2^ assessed by ultrasound; ^3^ assessed by pathology; LVSI = lymphovascular space invasion; ST = simple trachelectomy; RT = radical trachelectomy; SLNB = sentinel lymph node biopsy; LN = lymph nodes; MAC = macrometastasis; MIC = micrometastasis; ITC = isolated tumor cells; RT = radiotherapy; DOC = died of other cause; DOD = died of disease.

**Table 2 cancers-12-01438-t002:** Lymph node status. Combined results of SLN ultrastaging and pelvic non-SLN examination (*n* = 226).

SLN	Non SLN	Final LN Status	*n* (%)
Negative	Negative	Negative	188 (83.2%)
Negative	Positive (MAC)	Positive	2 (0.9%)
Positive MIC 1	Positive (MAC)	Positive	1 (0.4%)
Positive ITC 1 MIC 2 MAC 1 MAC 2	Positive ITC 1 MIC 2 MIC 1 MAC 2	Positive	6 (2.7%)
Positive ITC 7 MIC 14 MAC 8	Negative	Positive	29 (12.8%)

**Table 3 cancers-12-01438-t003:** Characteristics of patients with recurrence (*n* = 22).

No	Age	FST	Tumor Type	Stage pT	LN Status	LVSI	Largest Tumour Size	DSI	Type of Parametrectomy	Adjuvant Treatment	Disease Free Interval	Site of Recurrence	Current Status
**1**	33		SCC	1b1	0	Yes	20	1/3	C1	0	71	Comb	DOD
**2**	30	ART	A	1b1	0	0	23	1/3	C1	0	7	Pelvic	NED
**3**	53		A	1b1	ITC	0	34	2/3	C2	0	25	Comb	DOD
**4**	65		SCC	1b1	0	Yes	22	1/3	C1	0	28	Pelvic	DOD
**5**	37	ART	A	1b1	0	0	20	1/3	C1	0	8	Pelvic	NED
**6**	29	ART	A	1b1	0	0	10	1/3	C1	0	22	Pelvic	NED
**7**	62		SCC	1b1	0	Yes	22	3/3	C1	CombRT	69	Distant	DOD
**8**	46		AS	1b1	0	Yes	36	2/3	C2	0	4	Comb	DOD
**9**	25		SCC	1b2	MIC	Yes	55	3/3	C2	CHRT	16	Pelvic	DOD
**10**	41		AS	1b2	MIC	Yes	45	3/3	C2	CHRT	25	Distant	DOD
**11**	35		SCC	1b1	MAC	Yes	45	3/3	C2	CHRT	10	Comb	DOD
**12**	20	Cone	SCC	1b1	0	Yes	22	1/3	NA	0	7	Pelvic	DOD
**13**	65		SCC	1b2	MIC	Yes	45	3/3	C2	CombRT	21	Distant	DOD
**14**	30		SCC	1b1	0	0	26	2/3	C1	0	17	Pelvic	NED
**15**	32	Cone	A	1b1	0	0	13	1/3	NA	0	11	Comb	DOD
**16**	43		SCC	1b1	0	Yes	23	1/3	C2	0	14	Comb	DOD
**17**	29	Cone	SCC	1b1	0	0	25	3/3	NA	0	6	Pelvic	DOD
**18**	42		SCC	1b1	0	Yes	30	2/3	C2	0	7	Comb	DOD
**19**	34		SCC	1b2	MIC	Yes	68	3/3	C2	CHRT	3	Comb	DOD
**20**	44		A	1b1	MAC	Yes	25	2/3	C2	CHRT	6	Distant	DOD
**21**	61		SCC	2b	MAC	Yes	25	2/3	C2	CombRT	2	Comb	DOD
**22**	50		SCC	2b	MIC	Yes	32	3/3	C2	CHRT	26	Comb	DOD

A = adenocarcinoma; ART = abdominal radical trachelectomy; Comb = combined recurrence (pelvic plus distant); CHRT= concomitant chemoradiotherapy; CombRT = combined radiotherapy; DOD = died of disease; DSI = depth of stromal invasion; FST = fertility sparing treatment; ITC = isolated tumor cells; LVSI = lymphovascular space invasion; MAC = macrometastasis; MIC = micrometastasis; NA = not applicable; NED = no evidence of disease; SCC = squamous cell carcinoma.

**Table 4 cancers-12-01438-t004:** Significant parameters for the risk of recurrence from univariate analysis.

Predictor		Total *n* (*n* Recurrence)	HR (95% CI)	*p*-Value ^1^
Tumor type	Squamous	171 (15)	ref.	
	Adenocarcinoma	49 (5)	1.19 (0.43; 3.29)	0.731
	Adenosquamous	6 (2)	5.08 (1.15; 22.35)	0.032
LVSI	No	128 (7)	ref.	
	Yes	98 (15)	2.95 (1.20; 7.23)	0.018
Number of positive LN		226 (22)	1.50 (1.08; 2.09)	0.015
LN positivity, variant A	No	188 (13)	ref.	
	Yes (Any type)	38 (9)	3.71 (1.59; 8.69)	0.003
LN positivity, variant B	ITC, negative	196 (14)	ref.	
	MAC, MIC	30 (8)	4.03 (1.69; 9.62)	0.002
LN positivity, variant C	Negative	188 (13)	ref.	
	ITC	8 (1)	1.96 (0.26; 14.97)	0.518
	MAC	14 (3)	3.61 (1.03; 12.69)	0.046
	MIC	16 (5)	4.62 (1.65; 12.95)	0.004
Minimal TFD		196 (18)	0.87 (0.74; 1.03)	0.116
TFD binarized ^2^	> 3.45	65 (1)	ref.	
	≤ 3.45	131 (17)	9.00 (1.20; 67.63)	0.033
Tumor size binarized ^2^	≤ 33.5	151 (10)	ref.	
	> 33.5	75 (12)	2.56 (1.10; 5.94)	0.029
Adjuvant treatment	No	194 (14)	ref.	
	Yes	32 (8)	3.46 (1.45; 8.25)	0.005
Stage pT	1a	15 (0)	-	-
	1b1	157 (15)	ref.	
	≥1b2	54 (7)	1.37 (0.56; 3.36)	0.491

^1^ hazard ratios are computed using Cox proportional hazards model; ^2^ cut-off determined by ROC analysis, the criterion was the highest value of the sum of sensitivity and specificity; LVSI = lymphovascular space invasion; TFD = tumor free distance.

**Table 5 cancers-12-01438-t005:** Multivariate model for the risk of recurrence.

Predictor		OR (95% IS)	*p*-Value	HR (95% IS)	*p*-Value ^1^
Area = 0.799; *p* < 0.001					
Tumor type	Adenocarcinoma (ref. Squamous)	1.36 (0.33; 5.69)	0.670	1.24 (0.33; 4.61)	0.749
	Adenosquamous (ref. Squamous)	7.29 (0.86; 62.07)	0.069	4.86 (1.00; 23.61)	0.050
LVSI	Yes (ref. No)	2.13 (0.59; 7.74)	0.250	1.85 (0.56; 6.14)	0.318
Number of positive LN		0.82 (0.37; 1.82)	0.624	0.80 (0.39; 1.65)	0.546
LN positivity	MAC, MIC (ref. ITC, negative)	3.62 (0.46; 28.51)	0.222	3.56 (0.54; 23.63)	0.188
TFD binarized ^2^	≤3.45 (ref. >3.45)	5.27 (0.63; 43.92)	0.125	5.25 (0.65; 42.34)	0.119
Tumor size binarized ^2^	> 32.5 (ref. ≤ 32.5)	0.64 (0.18; 2.25)	0.486	0.72 (0.24; 2.17)	0.554
Adjuvant treatment	Yes (ref. No)	2.45 (0.48; 12.39)	0.279	1.78 (0.37; 8.57)	0.472

^1^ hazard ratios are computed using the Cox proportional hazards model; ^2^ cut-off determined by ROC analysis, the criterion was the highest value of the sum of sensitivity and specificity; LVSI = lymphovascular space invasion; TFD = tumor free distance.

**Table 6 cancers-12-01438-t006:** Overview of articles reported the impact of MIC on the prognosis.

Author (year)	No.	Stage	Tumor Type	SLNB	Adj. Tx (%)	mF/U (m)	LN Positivity				Recurrence Rate						Reported Impact of MIC on the Outcome
							% N1	%ITC	%MIC	%MAC	All	N0	N1 *	ITC	MIC	MAC	
Juretzka (2004)	49	IA2-IB2	SCC, A, AS, UD	**no**	**22**	**39**	8% (4/49)	n/a	8% (4/49)	n/a	10% (5/49)	6.7% (3/45)	4% (2/49)	n/a	50% (2/4)	n/a	**↑RecR**
Marchiole (2005)	52 (292)	IB1-IIB	SCC, A	no	15	122	n/a	11.5% (6/52)	23% (12/52)	n/a	8,9% (26/292)	n/a	n/a	n/a	21% (11/52)	n/a	**↑RecR** (RR = 2.44 (95% CI 1.58-3.78))
Fregnani (2006)	289	IB-IIA	SCC, A	no	13	102	17% (48/289)	2% (5/289)	2% (6/289)	13% (37/289)	14.9% (43/289)	n/a	n/a	n/a	n/a	n/a	**↑RecR** (RR N1mic = 3.2 (95% CI 1.1–9.6)), **↓5y DFS** (88.7% N0, 50% N1mic)
Horn (2008)	894	IB-IIB	SCC, A	no	31	82	31.4% (281/894)	n/a	6,5% (59/894)	23% (207/894)	17.8% (135/894)	n/a	n/a	n/a	n/a	n/a	**↓5y DFS** (91,4% for pN0, 69% pN1mic), **↓5y OS** (86,6% pN0, 63,8% pN1mic)
Cibula (2012)	645	IA-IIB	SCC, A, AS	yes	33	40	29.3% (189/645)	4.5% (29/645)	10.1% (65/645)	14.7% (95/645)	n/a	n/a	n/a	n/a	n/a	n/a	**↓OS** (HR = 6.86 (95% CI 2.09-22.61)), **→DFS**
Colturato (2015)	83	IB1-IIA	SCC, A, AS	no	0	60	n/a	7% (6/83) ^#^		n/a	18% (15/83)	18% (15/83)	n/a	27% (4/15) ^#^		n/a	**↑RecR** (OR = 11.73 (95% CI 1.57-87.8))
Stany (2015)	129	IA2-IB2	SCC, A, AS	no	23	70	n/a	20% (26/129) ^#^		n/a	8,5% (11/129)	n/a	n/a	18% (2/11) ^#^		n/a	**→3y DFS**
Guani (2019)	139	IA-IB1	SCC, A, AS	yes	19	36	15% (21/139)	4.3% (6/139)	5.7% (8/139)	5.7% (8/139)	9% (13/139)	9% (11/118)	10% (2/21)	0% (0/6)	14% (1/7)	12.5% (1/8)	**→3y DFS**
This study (2020)	226	IA-IIB	SCC, A, AS	yes	16	65	17% (38/226)	3.5% (8/226)	7% (16/226)	6% (14/226)	10% (22/226)	7% (13/188)	24% (9/38)	12.5% (1/8)	31% (5/16)	21% (3/14)	**↓DFS** (HR = 4.62 (95% CI 1.65-12.95)), **↓OS**

→ no impact, ↑ increased, ↓ decreased, A—adenocarcinoma, Adj. Tx—adjuvant treatment, AS—adenosquamous cancer, CI—confidence interval, DFS—disease-free survival, mF/U—median follow-up, HR—hazard ratio, LN—lymph node, N0—lymph node negative, N1—lymph node positive, ITC—isolated tumor cells, m—months, MAC—macrometastases, MIC—micrometastases, No.—number, OR—odds ratio, OS—overall survival, RecR—recurrence rate, RR—relative risk, SCC—squamous cell cancer, SLNB—sentinel lymph node biopsy, UD—undifferentiated; * MAC/MIC/ITC, ^#^ MIC + ITC.

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
