# Peer review of "Micrometastases in Sentinel Lymph Nodes Represent a Significant Negative Prognostic Factor in Early-Stage Cervical Cancer: A Single-Institutional Retrospective Cohort Study"

_cancers, 2020, doi:10.3390/cancers12061438_

Round 1

Reviewer 1 Report

The author has organized the data in detail and has done a lot of analysis. I believe it will be helpful for clinical work. I sugeest that the content figure1 and 2 could be displayed separately.  

Reviewer 2 Report

This is a retrospective analysis assessing the impact of MIC and MAC on prognosis of patients with early stage cervical cancer. The conclusions are interesting, even if "not really unexpected". The manuscript is well written but some minor english language revision should be performed. Most important thing is the statement by the authors (in the "conclusion" paragraph for example, that it is only a retrospective analysis, thus the low nomber of patients and the heterogeneity of the treatments performed (9 out of 22 patients with N+ disease performed chemoRT and 13 did not) colud affect the conclusions.

I recommend to reduce the "discussion", highlighting so only the most important feastures, avoiding to mention so many data.

Reviewer 3 Report

Remarks to the Author:

This study from Roman Kocian et al. aims to investigate the influence of the presence of MIC in the outcome of early-stage cervical cancer. The study is done on a relatively large cohort of patients, all treated in the same centre. The study is well explained and the methods as well as the results well described. I think, since the authors emphasized the importance of ultra-staging to be routinely done, it would be good to include in the study a comparison between what is currently routinely done in pathology and their ultra-staging protocol, to give a clear idea about the extra cost and labour it implies.

I also have some minors comments about the manuscript :

Line 27 : please develop SLN here

Line 75/76 : adjuvant therapy given to 32 patients : LN involvement 27, positive vaginal margin 2, parametrial involvement 8 : 27+2+8=37

Line 107 : I cannot find the number 2 in the table

Figure 1 and 2 :

It is hard to distinguish the different group, the different grey are to closed, especially on printed version

Can you include a short legend for both figure

Reviewer 4 Report

With high interest I reviewed the manuscript by Kocian et al. Their retrospective study aimed to assess the prognostic factor of (micro-)metastases in sentinel lymph nodes in early stage cervical cancer patients. 226 patients who underwent laparoscopic or open sentinel-guided lymph node dissection after injection of 99technetium and blue dye were enrolled in the presented analysis. In total, 38 women were lymph node positive (17%) including only 16 patients with micrometastases (macrometastases n=14, ITCs n=8). 22 recurrences occurred (median follow up 65 month). Furthermore, a not systematic review of the literature was integrated in the discussion section.  

In general, the manuscript is concise, well written and the analyses are sound.

However, some issues have to be considered:

The combination of an original article with a review of the literature is not common. A review should be presented as a high-quality (systematic) review in a separate article. The heading suggests that the article also includes a review. However, relevant literature is only discussed/an overview integrated in the discussion section. No information is given on the method of literature analysis in the method section. No results regarding the review can be found in the results section. Accordingly, the article is only a single institutional cohort study and does not include a review. The heading should be adjusted.

Information on ethical approval is missing. The authors must provide information on ethics, including patient informed consent.

Materials and Methods

If SLN were negative, full pelvic lymph node dissection was completed, expected In stage T1a/LVSI-negative patients. What is the rational here? Rather, it might make sense to extend lymphadenectomy, especially in patients with metastases positive SLNs, in order to remove lymph node metastases in secondary landing sites. Please explain further. How was the conventional lymphadenectomy performed?

The Sentinel approach should be explained in more detail (e.g., tracer volume, activity).

The intraoperative sentinel detection including the probe used should also be explained further. Which nodes have been defined as SLNs (e.g., +/- blue dfye)? Has a scintigraphy been performed preoperatively to visualize the SLNs?

Results

For the significance of the study, it is relevant in how many patients SLNs could be detected / resected. In addition, a differentiated analysis of positive lymph nodes (SLN / non-SLN?) should be provided and differentiated according to MIC / MAC / ITN.

Stratification taking into account the surgical technique (open vs. laparoscopic) should be added.

Table 1 shows that 93.8% of patients have had a lymphadenectomy. Does this only apply to conventional lymphadenectomy? It should be shown how many patients have received an SLNB + conventional lymphadenectomy or conventinal lymphadenectomy only.

Discussion

With regard to the influence of histopathological processing on the detection of lymph node metastases, the use of molecular methods should also be discussed.

Conclusions

In view of the small number of MIC-positive cases / recurrances examined and the heterogeneous operative procedure in the study presented, the conclusions drawn should be checked.

Round 2

Reviewer 2 Report

The authors have provided the requested revisions

Reviewer 4 Report

The authors handled all comments point by point. All necessary corrections were carried out and shortcomings were thus eliminated which has further increased the expressiveness of the manuscript.